# Serological Prevalence of Crimean–Congo Hemorrhagic Fever Virus Infection in Small Ruminants and Cattle in The Gambia

**DOI:** 10.3390/pathogens12060749

**Published:** 2023-05-23

**Authors:** Jerusha Matthews, Arss Secka, D. Scott McVey, Kimberly A. Dodd, Bonto Faburay

**Affiliations:** 1Foreign Animal Disease Diagnostic Laboratory, National Veterinary Services Laboratories, Animal and Plant Health Inspection Services, United States Department of Agriculture, Greenport, NY 11944, USA; jerusha.matthews@usda.gov; 2West Africa Livestock Innovation Center, Banjul PMB 14, The Gambia; seckaarss@gmail.com; 3School of Veterinary Medicine and Biomedical Sciences, University of Nebraska, Lincoln, NE 68933, USA; dmcvey2@unl.edu; 4Veterinary Diagnostic Laboratory, College of Veterinary Medicine, Michigan State University, East Lansing, MI 48824, USA; doddkimb@msu.edu

**Keywords:** Crimean–Congo hemorrhagic fever virus, serological prevalence, small ruminants, cattle, The Gambia

## Abstract

Crimean–Congo hemorrhagic fever virus (CCHFV) is a widely distributed tickborne zoonotic agent that infects a variety of host species. There is a lack of information on the true geographic distribution of the prevalence and risk of CCHFV in West Africa. A countrywide cross-sectional study involving 1413 extensively managed indigenous small ruminants and cattle at livestock sales markets and in village herds, respectively, was carried out in The Gambia. In sheep, an overall anti-CCHFV antibody prevalence of 18.9% (95% CI: 15.5–22.8%), goats 9.0% (95% CI: 6.7–11.7%), and cattle 59.9% (95% CI: 54.9–64.7%) was detected. Significant variation (*p* < 0.05) in the prevalence of anti-CCHFV antibodies at sites in the five administrative regions (sheep: 4.8–25.9%; goats: 1.8–17.1%) and three agroecological zones (sheep: 8.9–32.9%; goats: 4.1–18.0%) was also observed. Comparatively, higher anti-CCHFV antibody prevalence was detected in cattle (33.3–84.0%) compared to small ruminants (1.8–8.1%). This study represents the first countrywide investigation of the seroprevalence of CCHFV in The Gambia, and the results suggest potential circulation and endemicity of the virus in the country. These data provide critical information vital to the development of informed policies for the surveillance, diagnosis, and control of CCFHV infection in The Gambia and the region.

## 1. Introduction

Crimean–Congo hemorrhagic fever (CCHF) is a zoonotic, arboviral disease that presents asymptomatically in animals and can lead to systemic hemorrhagic symptoms in severe human cases [1]. Human infections often occur due to tick bites or direct contact with infected animals, and the disease is characterized by symptoms of fever, headaches, and myalgia. The case fatality ratios range from 5–40% [1,2], and the exposure risk is high for professionals in veterinary care, farmers, and abattoir workers [2,3].

The etiological agent of CCHF, Crimean–Congo hemorrhagic fever virus (CCHFV), is an enveloped, single-stranded, negative sense RNA virus belonging to the family *Nairoviridae* and genus *Orthonairovirus* [4]. The genome of CCHFV is tri-segmented, composed of small (S), medium (M), and large (L) segments [1]. The S segment encodes, in an overlapping open reading frame, the nonstructural protein (NSs) and nucleocapsid (N) protein, a highly immunogenic protein and a major target for serodiagnostic test development. The M segment encodes the glycoprotein precursor (GPC), the maturation of which yields the glycoproteins, Gn and Gc, and a non-structural M protein (NSm), and secreted non-structural proteins GP160, GP85, and GP38; and the L segment encodes the RNA-dependent RNA polymerase [5,6]. CCHFV is primarily transmitted by ixodid (hard-bodied) ticks in the genus *Hyalomma* [7]. Studies on ticks collected from hosts can indicate CCHFV circulation; however, additional data including host prevalence is needed to better understand vector competence. Such studies would more accurately establish the true prevalence and risk of CCHFV infection in a given geographic location [8]. The pathogenesis of CCHFV infection is poorly understood. In experimental infections, sheep and cattle develop only transient mild fever. Viremia levels and duration are relatively low and short and detectable by reverse-transcriptase-PCR. Enzyme-linked immunosorbent assays (ELISA) can detect IgG antibodies for the remainder of the life of the animal [9]. 

In West Africa, previous studies established CCHFV prevalence in domestic livestock and suggest ruminants as suitable indicator species to determine human risk [10,11,12,13,14]. Serological studies of CCHFV infections in domestic livestock and humans, including detection of infection in ticks, in Senegal and Nigeria have been reported [7,14,15,16,17]; however, the true burden of CCHF endemicity in large parts of West Africa remains undetermined. The current study represents the first of its kind in The Gambia, aimed at investigating the seroprevalence and risk of CCHFV infection in indigenous livestock populations and potentially for in-contact human populations in the country.

In this study, a cross-sectional sampling was carried out to assess CCHFV seroprevalence in areas with high human–livestock interaction and among village cattle herds. The study involved sampling small ruminants, sheep and goats, located at various livestock aggregation centers across the five administrative regions and three agroecological zones of The Gambia, as well as extensively managed cattle, exposed to natural field tick challenge, in herds located in rural villages. The study provides critically important information for the first time on the seroprevalence of CCHFV in The Gambia and highlights potential endemicity of the virus in the country. 

## 2. Materials and Methods

### 2.1. Study Sites and Sample Population

A cross-sectional sampling was carried out between April and May 2021 in small ruminants, sheep (Djallonke, Sahelian and crosses) and goats (West African dwarf, Sahelian and crosses), and cattle (N’Dama breed) at selected sites in The Gambia. For small ruminants, samples were collected from indigenous traditionally managed sheep (n = 470) and goats (n = 544), aged 6 months and above, at livestock aggregation centers consisting mainly of livestock markets and a breeding facility located in the 5 administrative regions (Figure 1) and 3 agroecological zones of The Gambia described below. Animals located at these sampling sites are often in close interaction with human populations. In the Western Region (WR), samples were collected from Abuko abattoir/livestock market and the Brikama livestock market, the Lower River Region (LRR) from the West Africa Livestock Innovation Center (WALIC) Keneba station and the Soma livestock market, the North Bank Region (NBR) from the Farafenni livestock market, Central River Region (CRR) from Brikamaba livestock market and Sololo WALIC station, and Upper River Region (URR) from the Basse livestock market. Most small ruminants at these markets have been sourced from surrounding villages, where they have been maintained under traditional husbandry and management systems and exposed to field ticks without acaricide treatment. A sizeable number of small ruminants at the Soma and Farafenni livestock markets are typically sourced from locations in Northern Senegal in a predominantly arid region with comparatively minimal or limited exposure to vector tick challenge. 

For cattle, samples were collected from the indigenous N’Dama breed (n = 399) from 10 village herds located in the Lower River Region in February 2021. The cattle were managed in herds and average sample size was 40 (range n = 30–65); the animals were maintained under a traditional husbandry free-range system, exposed to sustained tick challenge without acaricide treatment or tick control. Samples were collected from cattle of various age categories, ranging from young adults (under 4 years old) and adults (more than 4 years old). Georeferenced coordinates were recorded for all sampling locations. All serum samples were subjected to heat inactivation at 56 °C for 2 h prior to shipment to the National Veterinary Services Laboratories’ Foreign Animal Disease Diagnostic Laboratory (NVSL/FADDL) for testing for antibodies against CCHFV infection. 

The agroecological zones consisted of the Sudano-Guinea (SG), Western Sudano-Sahelian (WSS), and Eastern Sudano-Sahelian (ESS) zones, as described previously [18]. Briefly, sampling sites in the SG zone (13.43 N, 16.72 W) were located within the 900 and 1200 mm of rainfall isohytes with maximum daily temperatures ranging from 26 to 32 °C. The vegetation is savannah-woodland or woodland in certain areas. In some areas around the coast, the vegetation is characterized by humid tropical forest vegetation. Sampling sites in the WSS zone (13.20 N, 16.01 W) experienced an average of 800 mm of rainfall with maximum daily temperatures ranging from 28 to 38 °C. The vegetation is composed of degraded savannah woodland interspersed with natural unimproved grasslands. The agroecology of sampling sites in the ESS zone (13.27 N, 14.40 W) typically received on average 700 mm of rainfall isohytes with maximum temperatures ranging from 30 to 40 °C. The vegetation is mainly open savannah and riparian woodland towards the river. 

### 2.2. Serological Analysis

Serum samples were tested in duplicate following manufacturers’ instructions using ID Screen CCHF double antigen multi-species ELISA kits (IDVet, Grabels, France). The ELISA kits have high sensitivity (98.9%; 95% CI: 96.8–99.8%) and specificity (100%; 95% CI: 99.8–100%) for detection of antibodies against CCHFV across different species simultaneously [19]. The tests were carried out according to manufacturer’s instructions. Briefly, 96-well microtiter plates were precoated with recombinant CCHFV nucleoprotein antigen. Test sera including negative and positive controls were added and incubated at 21 °C (±5 °C) for 45 min. After washing, recombinant nucleoprotein conjugated to horseradish peroxidase (HRP) was added to each well and incubated at 21 °C (±5 °C) for 30 min. After washing, substrate solution was added to each well and incubated in the dark at 21 °C (±5 °C) for 15 min. The reaction was stopped with a stop solution and optical density (OD) was measured at 450 nm. Samples with sample to positive ratio (S/P %) greater than 30% were determined positive. 

### 2.3. Statistical Analysis

All statistical analyses were performed using STATA statistical software version 15 (StataCorp LLC, TX). Seroprevalence of infection was calculated by determining the percentage of positive samples within each variable category. Differences in seroprevalence across variables were determined using odds ratio and chi-square tests. Univariate logistic regression models were used to test the association of CCHFV seroprevalence with study site, region, and agroecological zone for sheep and goats. For cattle, univariate and multivariate logistic regression models were used to test the association of CCHFV seroprevalence with study site, age, and sex. In the univariate analysis, variables that were associated with the outcomes at a significant level of 20% (*p* ≤ 0.2) were considered statistically significant. A 95% confidence interval and a significance level of 5% were used to determine statistical significance of the multivariate model. The graphs were created using GraphPad Prism Version 8.

## 3. Results

### 3.1. CCHFV Seroprevalence in Sheep

Of the 470 sheep samples tested across study sites, 18.9% (*n* = 89; 95% CI: 15.5–22.8%) were seropositive for antibodies against CCHFV (Table 1). Seroprevalence was highest at the Sololo WALIC station (38.1%; 95% CI: 26.1–51.2%), followed by Abuko abattoir (29.6%; 95% CI: 21.8–38.4%), Basse livestock market (23.5%; 95% CI: 10.7–41.2%), and Brikamaba livestock market (11.3%; 95% CI: 4.2–23.0%). Seroprevalences at the other study sites were comparatively lower: Soma livestock market (9.1%; 95% CI: 0.2–41.3%), Keneba WALIC station (8.5%; 95% CI: 4.3–14.7%), Farafenni livestock market (4.8%; 95% CI: 0.1–23.8%), and Brikama livestock market (2.9%; 95% CI: 0.1–15.3%). In the univariate analysis, there were lower odds of anti-CCHFV seropositivity in the Brikama livestock market (OR = 0.07, *p* = 0.011, 95% CI: 0.01–0.54), Brikamaba livestock market (OR = 0.30, *p* = 0.012, 95% CI: 0.11–0.77), Farafenni livestock market (OR = 0.11, *p* = 0.041, 95% CI: 0.01–0.91), and Keneba WALIC station (OR = 0.22, *p* < 0.001, 95% CI: 0.10–0.45) compared to the major livestock market at the Abuko abattoir. Overall, CCHFV seropositivity detected at the Abuko abattoir (29.6%) was significantly higher than seropositivity detected at the Brikama livestock market (2.9%), Brikamaba livestock market (11.3%), Farafenni livestock market (4.8%), and Keneba WALIC station (8.5%) (*p* < 0.05; Table 1). 

Overall prevalence of CCHF antibodies was determined cumulatively for sites in each of the 5 administrative regions, with similar levels of seropositivity detected in the Central River (25.9%; 95% CI: 18.2–34.8%), Upper River (23.5%; 95% CI: 10.7–41.2%), and Western (23.9%; 95% CI: 17.5–31.3%). Seroprevalences were significantly lower in the Lower River (8.6%; 95% CI: 4.5–14.5%; *p* < 0.05) and North Bank (4.8%; 95% CI: 0.1–23.8%; *p* < 0.05). There were lower odds of anti-CCHFV antibody seropositivity in the Lower River (OR = 0.26, *p* < 0.001, 95% CI: 0.13–0.55) compared to Central River, a region with the country’s highest livestock population and high abundance of the major tick vector, *Hyalomma* spp. Overall, significantly higher seropositivity was detected at sites in the Central River Region (32.9%) than at sites in the Lower River Region (8.6%) (*p* < 0.001). 

Of the three agroecological zones (AEZ) in The Gambia, overall seroprevalences of CCHFV antibodies were comparatively higher in the Eastern Sudano-Sahelian (32.9%; 95% CI: 23.8–43.3%) and Sudano-Guinean (23.9%; 95% CI: 17.5–31.3%) than in the Western Sudano-Sahelian zone (8.9%; 95% CI: 5.4–13.5%). There were lower odds of anti-CCHFV seropositivity in the Western Sudano-Sahelian (OR = 0.19, *p* < 0.001, 95% CI: 0.10–0.37) compared to Eastern Sudano-Sahelian. Overall, the prevalence of CCHF antibodies was significantly higher at sites in the Eastern Sudano Sahelian zone (32.9%) than sites in the Western Sudano Sahelian zone (8.9%) (*p* < 0.001). 

### 3.2. CCHFV Seroprevalence in Goats

Of the 544 goats tested across study sites, 9.0% (n = 49; 95% CI: 6.7–11.7%) were seropositive for anti-CCHFV antibodies (Table 2). The highest seroprevalence was detected at Sololo WALIC station (31.2%; 95% CI: 16.1–50.0%), followed by Abuko abattoir (19.6%; 95% CI: 9.8–33.1%) and Basse livestock market (12.6%; 95% CI: 6.2–22.0%). Lower seroprevalences were detected at the remaining sites: Brikama livestock market (7.7%; 95% CI: 3.1–15.2%), Brikamaba livestock market (5.3%; 95% CI: 0.6–17.7%), Keneba WALIC station (4.6%; 95% CI: 1.8–9.2%), Soma livestock market (4.4%; 95% CI: 0.5–15.1%), and Farafenni livestock market (1.8%; 95% CI:0.1–9.5%). In a univariate analysis, the odds of anti-CCHFV seropositivity were lower in the Farafenni livestock market (OR = 0.07, *p* = 0.015, 95% CI: 0.01–0.61), Keneba WALIC station (OR = 0.19, *p* = 0.002, 95% CI: 0.07–0.55), Soma livestock market (OR = 0.19, *p* = 0.039, 95% CI: 0.03–0.92), and Brikamaba livestock market (OR 0.22, *p* = 0.067, 95% CI: 0.04–1.10) compared to Abuko abattoir. Overall, CCHFV seroprevalence detected at the Abuko abattoir (19.6%) was significantly higher than seroprevalence detected at Brikama livestock market (7.7%), Farafenni livestock market (1.8%), and Keneba WALIC station (4.6%) and Soma livestock market (4.4%) (*p* < 0.05; Table 2). 

Overall, prevalence of CCHFV antibodies in goats was determined for the five administrative regions, with the highest seroprevalence detected in Central River (17.1%; 95% CI: 9.1–28.0%). Similar levels of seropositivity were detected in Upper River (12.7%; 95% CI: 6.2–22.0%) and Western (11.9%; 95% CI: 7.1–18.5%). Seroprevalences were comparatively lower for Lower River (4.6%; 95% CI: 2.1–8.5%) and North Bank (1.8%; 95% CI: 0.1–9.5%). Odds of anti-CCHFV seropositivity were lower in Lower River (OR = 0.23, *p* = 0.002, 95% CI: 0.09–0.57) compared to Central River. Overall, significantly higher seroprevalence was detected at sites in the Central River Region (17.1%) than sites in the Lower River Region (4.6%) and North Bank Region (1.8%) (*p* < 0.001). For agroecological zones, CCHFV seroprevalence was comparatively higher in goats at sites in the Eastern Sudano-Sahelian (18.0%; 95% CI: 11.3–26.4%) and Sudano-Guinean (11.9%; 95% CI: 7.1–18.5%) zones. Significantly lower seroprevalence was detected in the Western Sudano-Sahelian zone (4.1%; 95% CI: 2.1–7.1%, *p* < 0.05). There were lower odds of anti-CCHFV antibody seropositivity in Western Sudano-Sahelian (OR = 0.19, *p* < 0.001, 95% CI: 0.09–0.41) compared to the Eastern Sudano Sahelian (Table 2). Overall, the prevalence of CCHF antibodies was significantly higher at sites in the Eastern Sudano Sahelian zone (18.0%) than sites in the Western Sudano Sahelian zone (4.1%) (*p* < 0.001). 

### 3.3. CCHFV Seropositivity in the Administrative Regions and Agroecological Zones

To further compare differences in CCHFV seroprevalence for sites in the different administrative regions, the proportion of CCHFV seropositive animals in the different regions for both sheep and goats was compared. Overall, sites in CRR, WR, and URR accounted for the highest proportion of CCHFV seropositive animals in sheep (30%, 27.5%, and 27.2%, respectively). For goats, the highest proportion of CCHFV seropositivity were detected at sites in CRR, URR, and WR (36%, 26%, and 25%, respectively) (Figure 2A,B). The lowest proportion of CCHFV seropositivity for both sheep and goats occurred at sites in LRR and NBR (9.8% and 5.5% sheep, 9% and 4% goats, respectively). To compare differences in CCHFV seropositivity for sites located in the different AEZs, we compared the proportion of CCHFV seropositivity between AEZs for both sheep and goats. Sites located within the ESS and SG accounted for the highest proportion of CCHFV seropositive animals in both sheep and goats (49.0% and 34.0% sheep, 52.9% and 35.0% goats, respectively) (Figure 2C,D). WSS accounted for the lowest proportion of CCHFV seropositivity for both sheep and goats (17% and 12%, respectively) (Figure 2C,D).

### 3.4. CCHFV Seroprevalence in Cattle

Of the 399 cattle tested across study sites or villages, 59.9% (n = 239; 95% CI: 54.9–64.7%) were seropositive for anti-CCHFV antibodies (Table 3). Seroprevalence was highest at Wudeba (84.0%; 95% CI: 70.9–92.8%), followed by Bajana (80.0%; 95% CI: 61.4–92.3%), Jiffarong (80.0%; 95% CI: 61.4–92.3%), Tankular (76.7%; 95% CI: 57.7–90.1%), Kuli Kunda (73.3%; 95% CI: 54.1–87.7%), and Niorro Jataba (73.5%; 95% CI: 58.9–85.1%). Seroprevalences were comparatively lower in cattle in Jali (43.3%; 95% CI: 25.5–62.6%), Dumbuto (43.1%; 95% CI: 30.8–55.9%), Sankandi (33.3%; 95% CI: 20.0–48.9%), and Bateling (30.0%; 95% CI: 16.6–46.5%). In a univariate analysis, there were lower odds of anti-CCHFV seropositivity in Bateling (OR = 0.10, *p* < 0.0001, 95% CI: 0.03–0.32), Dumbuto (OR = 0.18, *p* = 0.001, 95% CI: 0.07–0.52), Jali (OR = 0.19, *p* = 0.005, 95% CI: 0.06–0.60), and Sankandi (OR = 0.12, *p* < 0.001, 95% CI: 0.04–0.37) compared to the village of Bajana. After adjusting for all tested variables in a multivariate analysis, similar pattern for low odds of seropositivity was observed for the villages mentioned above compared to Bajana (Table 3). 

Seroprevalence was highest in cattle over the age of four years (63.1%; 95% CI: 56.1–69.7%); cattle under the age of four years were comparatively lower (56.5%; 95% CI: 49.1–63.6%). To determine the possible effect of sex on CCHFV seroprevalence in indigenous cattle exposed to natural field tick challenge, we compared the seroprevalence of infection between male and female cattle. Higher seroprevalence was detected in female cattle (64.2%; 95% CI: 58.5–69.6) compared to male (46.4%; 95% CI: 36.2–56.8%), with male cattle being less likely to be seropositive (unadjusted OR = 0.48, *p* < 0.002, 95% CI: 0.30–0.76) (adjusted OR = 0.31, *p* < 0.001, 95% CI: 0.18–0.54) compared to females (Figure 3). 

To examine possible differences in CCHFV seroprevalence among the ruminant livestock species, comparison was carried out in LRR, where all species, sheep, goats, and cattle, were sampled in the region. CCHFV seroprevalence in the LRR was not very different between goats (4.6%; 95% CI: 2.1–8.5%) and sheep (8.6%; 95% CI: 4.5–14.5%). Significantly higher CCHFV seroprevalence was detected in cattle (59.9%; 95% CI: 54.9–64.7%) compared to small ruminants, sheep and goats, in this administrative region (Figure 4).

## 4. Discussion

Crimean–Congo hemorrhagic fever is the most widespread viral tick-borne disease globally, having been reported in parts of Africa, Asia, and Europe [15,20]. Although the disease is endemic in Africa, few studies have been carried out to accurately map the true geographic distribution and risk of CCHF on the continent [20,21]. Despite previous studies describing the prevalence of CCHFV infection in Senegal, a close geographic neighbor of The Gambia, this study represents the first to describe the seroprevalence of CCHFV infection in domestic ruminant populations, small ruminants and cattle, across the five administrative regions encompassing the three major agroecological zones of The Gambia. Samples were collected from sheep and goats at major livestock aggregation points, primarily livestock markets, as well as livestock breeding facilities, where animals are located near population centers and in proximity to human populations, thus presenting significant risk of zoonotic disease transmission. Samples from cattle were collected from extensively managed village herds that were exposed to sustained field tick challenge and potential virus transmission by the major vector, *Hyalomma* spp. In sheep and goats, the seroprevalence of infection was highest at sites in the CRR, URR, and WR (Table 1 and Table 2). Significantly higher seroprevalences (*p* < 0.05) were detected at the Sololo WALIC livestock facility (38.1% sheep, 31.2% goats) and Abuko Abattoir (29.6% sheep, 19.6% goats) located in CRR in the ESS zone and WR in the SG zone, respectively. The latter region is host to the largest livestock market in the country receiving small ruminants from different geographic locations, including regions as far as southern and northern Senegal, whereas the former is host to a large, small ruminant breeding facility, receiving replenishment stock from traditionally reared animals from surrounding villages and beyond. The comparatively high CCHFV seroprevalence detected at these sites suggests high risk of CCHFV transmission to humans interacting with or in proximity to animals and suggests that animals at these sites may have been sourced from locations with high risk of CCHFV infection. In neighboring Senegal, an anti-CCHFV IgG antibody prevalence of 10.2% was detected in humans living in proximity to sheep exhibiting an anti-CCHFV IgG antibody prevalence of 38.42% [7]. Similarly, in Pakistan, a 2.7% CCHFV seroprevalence was detected in humans that live in close proximity to domestic ruminants that exhibit a CCHFV seroprevalence of 36.2% [22]. Further studies are necessary to establish a potential epidemiological link to these locations for potential exposure and infection in livestock and humans. Interestingly, the Brikama livestock market located in the WR demonstrated comparatively low seroprevalence in both sheep and goats, suggesting animals at this at this facility have been sourced from regions or locations with comparatively lower risk of CCHFV infection or transmission. This may further explain the low seroprevalence of CCHFV infection detected in small ruminants at the Farafenni and Soma livestock markets in the NBR and LRR, respectively. Through the towns of Soma and Farafenni, located in LRR and NBR, respectively, runs a major trading route for livestock that are most predominantly sourced from northern Senegal, a place with an arid climate and close to the southern limit of the Sahara Desert, characterized by comparatively low abundance of ixodid tick species, including the major vector, *Hyalomma* spp. [23]. 

Overall, significant variations in CCHFV seroprevalence have been detected in animals at various study sites. These sites are in geographic locations or agroecological zones that differ in annual precipitation and temperature [18], which may impact tick abundance [24]. Despite that, *Hyalomma* spp. have been described as primary vectors for CCHFV, and other tick species such as *Rhipicephalus* and *Amblyomma* spp. have been implicated in the transmission of CCHFV [12,20,25,26]; indeed, studies in The Gambia have demonstrated the abundance of these tick species at these AEZ regions [27,28]. The serological prevalence in domestic ruminants described in this study are comparable to prevalence estimates reported in domestic ruminants elsewhere in West Africa [7,13,14,17]. The anti-CCHFV antibodies in small ruminants suggests the risk of CCHFV infection for humans at the broader study areas or regions, but as well to individuals in close contact with these animals at livestock sales markets. While this information is important in describing the potential risk for CCHF at these study sites, further studies examining the seroprevalence and risk of infection in village small ruminant flocks will provide additional and specific information vital to determining the true risk of CCHFV infection for resident local communities. On the other hand, the seroprevalence of CCHFV infection in cattle described in this study may broadly reflect regional prevalence and the potential risk of CCHF for local communities, as the samples were collected from resident indigenous cattle maintained in village herds that were exposed to sustained field tick challenge and potential vector transmission. Indeed, comparable levels of CCHFV seroprevalence have been reported in domestic ruminants elsewhere in Africa. A seroprevalence of 66% in cattle in Mali, 69% in cattle, and 15% in sheep and goats in Mauritania, 57.7% in cattle in Niger; and as high as 98.8% and 16.65% in cattle and sheep, respectively, in Cameroon, have been reported [11,29,30,31]. Comparatively high CCHFV seroprevalence has been detected in cattle (range 30–84%), which was significantly higher than in small ruminants, sheep (2.9–38.1%) and goats (range 1.8–31.2%). These results agree with similar findings of higher prevalence of anti-CCHFV antibodies in cattle compared to sheep and goats [11,13,17,32,33], which could be attributed to a higher level of vector tick challenge resulting from higher tick infestation of cattle than small ruminants. Additionally, significantly (*p* < 0.001) higher prevalence of anti-CCHFV antibodies was detected in female cattle (64.2%) than in male cattle (46.4%). Interestingly, there are reports describing higher susceptibility of female cattle to tick infestation than male cohorts, with specific studies describing higher tick burden in female Pakistani and Colombian cattle than in their male cohorts [24,34]. These studies suggest female cattle may be exposed to potential higher infective tick challenge and could explain the higher anti-CCHFV antibody prevalence in the female cohort. Studies will be required in the future to confirm or refute this hypothesis. 

CCHFV infection in humans often is characterized by sudden onset of illness with a high-grade fever of over 38.5 °C for more than 72 h and less than 10 days, which symptoms, especially acute febrile illness, present frequently among residents and communities in The Gambia. Considering the high seroprevalence of CCHFV infection in domestic ruminants, sheep, goats, and cattle, it is reasonable to suggest that zoonotic transmission to humans that are in close contact with livestock is occurring, and the febrile symptoms may be misdiagnosed for malaria. In the 2008 malaria season in West Africa, only 11% of the febrile illnesses detected during a follow-up study in children were due to malaria [7], suggesting substantial misdiagnosis of malaria with treatable or preventable bacterial or arboviral infections [35]. This is highly likely, given that in The Gambia, peak tick abundance occurs during the rainy season (July to September), including high nymphal tick infestation of small ruminant populations in the early dry season (November to March) [36], with potential for increased risk of CCHFV transmission. This period of high tick infestation coincides with heightened mosquito abundance, further increasing the likelihood of misdiagnosis of symptoms of CCHFV infection with malaria. The endemicity and potential risk of CCHFV across regions in West Africa is highlighted by reports of localized outbreaks of the disease in humans in Senegal and Mauritania [37,38]. Considering the likelihood that the presence of antibodies against CCHFV in livestock may correlate with infection in humans [7], further studies are needed to investigate human infections, in a One Health approach, to aid in establishing risk and correlation between livestock infections and human infections in these areas. Information derived from such a study shall provide a more detailed and comprehensive assessment of the risk of CCHFV in The Gambia. 

## 5. Conclusions

In conclusion, this is the first study in The Gambia describing the prevalence of anti-CCHFV antibodies in domestic ruminants, of which the results suggest the circulation and endemicity of the virus in The Gambia. The ELISA test utilized in this study detects both IgG and IgM antibodies, with the latter suggesting more recent infection. This coupled with the high seroprevalence in small ruminants at livestock markets located in population agglomeration centers, where animals are in proximity or interact very closely with humans, suggest the potential for zoonotic transmission of the CCHFV in local communities. Future studies that identify acute infections in at-risk human populations as well as febrile patients via detection of pathogen nucleic acid and /or IgM antibodies need to be conducted to demonstrate circulation of the virus and a more accurate risk for zoonotic disease transmission. The significantly high level of seroprevalence detected in local cattle corroborates reports of cattle being good sentinels and important reservoirs for CCHFV infection and transmission in endemic regions and presents high occupational risk of zoonotic transmission to local herdsmen, attending veterinary care personnel, and abattoir workers. The results of this study will provide an important basis for informing national policy on the surveillance, diagnosis, and control of CCHFV infection in The Gambia as well as regionally in West Africa. 

## Figures and Tables

**Figure 1 pathogens-12-00749-f001:**
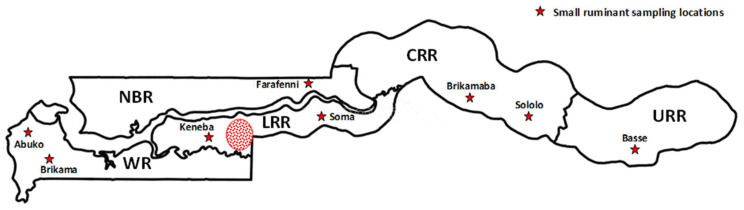
Map of The Gambia showing the sampling locations in the various administrative regions (WR = Western region, NBR = North Bank region, LRR = Lower River region, CRR = Central River region, URR = Upper River region). The red circle denotes a cluster of villages in LRR where samples from cattle were collected.

**Figure 2 pathogens-12-00749-f002:**
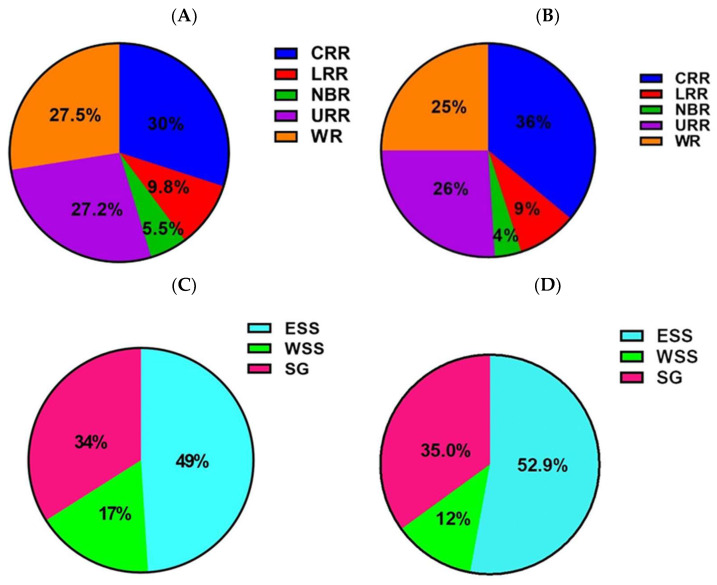
Proportion of overall CCHFV seropositivity detected in sheep (**A**,**C**) and goats (**B**,**D**) accounted for in each of the administrative regions (upper panel) or agroecological zones (lower panel).

**Figure 3 pathogens-12-00749-f003:**
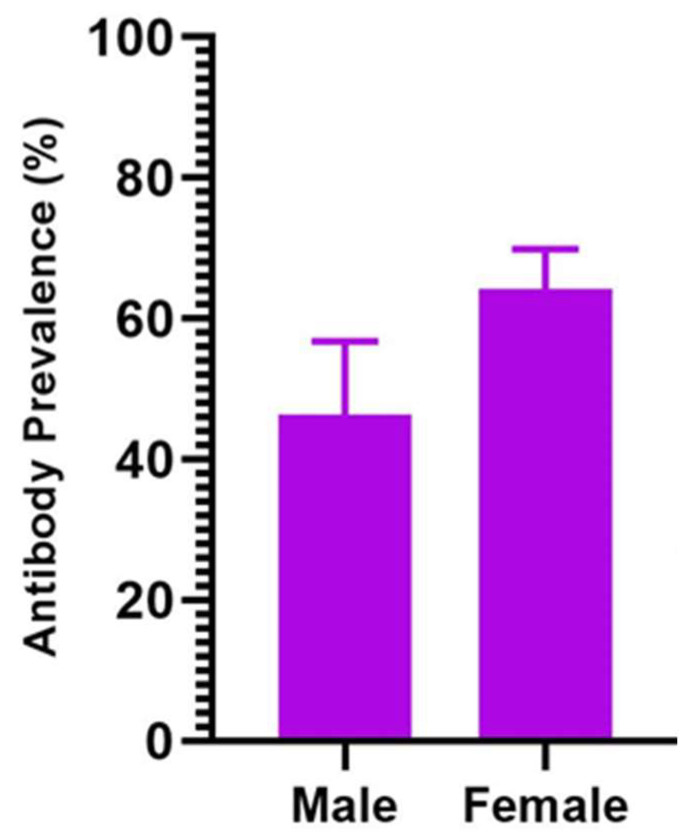
Prevalence of anti-CCHFV antibodies detected in male and female cattle populations. A significantly higher prevalence of antibodies was detected in female cattle (*p* < 0.001).

**Figure 4 pathogens-12-00749-f004:**
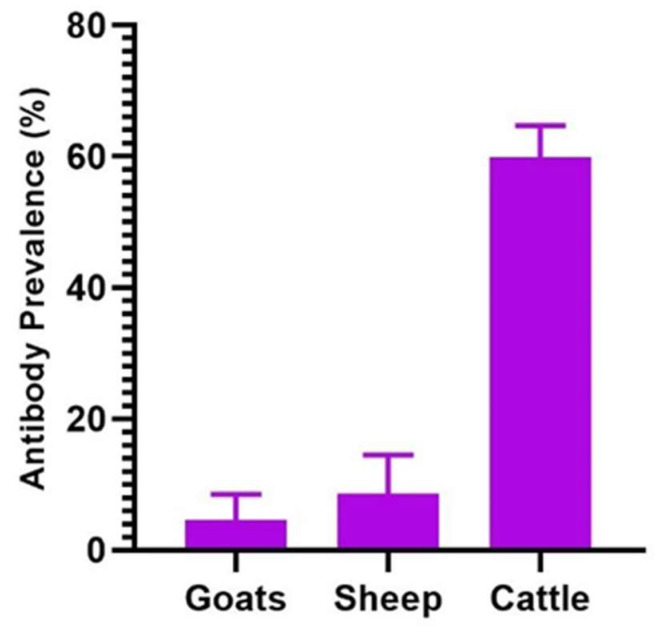
Prevalence of anti-CCHFV antibodies in goats, sheep, and cattle at study sites in the Lower River Region. Cattle exhibit significantly higher antibody prevalence than either sheep or goats (*p* < 0.001).

**Table 1 pathogens-12-00749-t001:** Seroprevalence of Crimean–Congo hemorrhagic fever in sheep across different regions in The Gambia, 2021.

Variable	* n/N	Seroprevalence % (95% CI)	Odds Ratio (95% CI)	*p*-Value
Overall	89/470	18.9 (15.5, 22.8)		
Site				
Abuko abattoir	37/125	29.6 (21.8, 38.4)	REF	
Basse livestock market	8/34	23.5 (10.7, 41.2)	0.73 (0.30, 1.76)	0.487
Brikama livestock market	1/34	2.9 (0.1, 15.3)	0.07 (0.01, 0.54)	0.011
Brikamaba livestock market	6/53	11.3 (4.2, 23.0)	0.30 (0.11, 0.77)	0.012
Farafenni livestock market	1/21	4.8 (0.1, 23.8)	0.11 (0.01, 0.91)	0.041
Keneba WALIC station	11/129	8.5 (4.3, 14.7)	0.22 (0.10, 0.45)	<0.001
Soma livestock market	1/11	9.1 (0.2, 41.3)	0.23 (0.03, 1.92)	0.178
Sololo WALIC station	24/63	38.1 (26.1, 51.2)	1.46 (0.77, 2.76)	0.241
Region				
Central River	30/116	25.9 (18.2, 34.8)	REF	
Lower River	12/140	8.6 (4.5, 14.5)	0.26 (0.13, 0.55)	<0.001
North Bank	1/21	4.8 (0.1, 23.8)	0.14 (0.02, 1.11)	0.063
Upper River	8/34	23.5 (10.7, 41.2)	0.88 (0.36, 2.15)	0.783
Western	38/159	23.9 (17.5, 31.3)	0.90 (0.51, 1.56)	0.710
Agroecological zone				
Eastern Sudano Sahelian	32/97	32.9 (23.8, 43.3)	REF	
Western Sudano Sahelian	19/214	8.9 (5.4, 13.5)	0.19 (0.10, 0.37)	<0.001
Sudano Guinean	38/159	23.9 (17.5, 31.3)	0.63 (0.36, 1.11)	0.115

***** n = Number positive for Crimean–Congo Hemorrhagic Fever Virus; N = Total sample examined; CI = Confidence Interval.

**Table 2 pathogens-12-00749-t002:** Seroprevalence of Crimean–Congo hemorrhagic fever in goats across different regions in The Gambia, 2021.

Variable	* n/N	Seroprevalence % (95% CI)	Odds Ratio (95% CI)	*p*-Value
Overall	49/544	9.0 (6.7, 11.7)		
Site				
Abuko abattoir	10/51	19.6 (9.8, 33.1)	REF	
Basse livestock market	10/79	12.6 (6.2, 22.0)	0.59 (0.22, 1.55)	0.287
Brikama livestock market	7/91	7.7 (3.1, 15.2)	0.34 (0.12, 0.96)	0.042
Brikamaba livestock market	2/38	5.3 (0.6, 17.7)	0.22 (0.04, 1.10)	0.067
Farafenni livestock market	1/56	1.8 (0.1, 9.5)	0.07 (0.01, 0.61)	0.015
Keneba WALIC station	7/152	4.6 (1.8, 9.2)	0.19 (0.07, 0.55)	0.002
Soma livestock market	2/45	4.4 (0.5, 15.1)	0.19 (0.03, 0.92)	0.039
Sololo WALIC station	10/32	31.2 (16.1, 50.0)	1.86 (0.67, 5.15)	0.231
Region				
Central River	12/70	17.1 (9.1, 28.0)	REF	
Lower River	9/197	4.6 (2.1, 8.5)	0.23 (0.09, 0.57)	0.002
North Bank	1/56	1.8 (0.1, 9.5)	0.08 (0.01, 0.69)	0.021
Upper River	10/79	12.7 (6.2, 22.0)	0.70 (0.28, 1.73)	0.443
Western	17/142	11.9 (7.1, 18.5)	0.65 (0.29, 1.46)	0.305
AEZ				
Eastern Sudano Sahelian	20/111	18.0 (11.3, 26.4)	REF	
Western Sudano-Sahelian	12/291	4.1 (2.1, 7.1)	0.19 (0.09, 0.41)	<0.001
Sudano-Guinean	17/142	11.9 (7.1, 18.5)	0.61 (0.30, 1.24)	0.179

***** n = Number positive for Crimean–Congo Hemorrhagic Fever Virus; N = Total sample examined; CI = Confidence Interval.

**Table 3 pathogens-12-00749-t003:** Seroprevalence of Crimean–Congo hemorrhagic fever in cattle within the Lower River region of The Gambia, 2021.

Variable	* n/N	Seroprevalence % (95% CI)	Unadjusted Odds Ratio (95% CI)	*p*-Value	Adjusted Odds Ratio (95% CI)	*p*-Value
Cattle seroprevalence	239/399	59.9 (54.9, 64.7)				
Site						
Bajana	24/30	80.0 (61.4, 92.3)	REF			
Bateling	12/40	30.0 (16.6, 46.5)	0.10 (0.03, 0.32)	<0.001	0.10 (0.03, 0.32)	<0.001
Dumbuto	28/65	43.1 (30.8, 55.9)	0.18 (0.07, 0.52)	0.001	0.20 (0.07, 0.57)	0.003
Jali	13/30	43.3 (25.5, 62.6)	0.19 (0.06, 0.60)	0.005	0.19 (0.06, 0.64)	0.007
Jiffarong	24/30	80.0 (61.4, 92.3)	1.00 (0.28, 3.54)	1.000	1.26 (0.34, 4.62)	0.727
Kuli Kunda	22/30	73.3 (54.1, 87.7)	0.68 (0.21, 2.29)	0.543	0.74 (0.21, 2.54)	0.635
Niorro Jataba	36/49	73.5 (58.9, 85.1)	0.69 (0.23, 2.07)	0.511	0.86 (0.27, 2.65)	0.794
Sankandi	15/45	33.3 (20.0, 48.9)	0.12 (0.04, 0.37)	<0.001	0.12 (0.03, 0.36)	<0.001
Tankular	23/30	76.7 (57.7, 90.1)	0.82 (0.23, 2.81)	0.754	1.02 (0.28, 3.63)	0.971
Wudeba	42/50	84.0 (70.9, 92.8)	1.31 (0.40, 4.23)	0.649	1.72 (0.51, 5.75)	0.377
Age group						
≤4 years	109/193	56.5 (49.1, 63.6)	0.76 (0.50, 1.13)	0.177	0.78 (0.49, 1.23)	0.291
>4 years	130/206	63.1 (56.1, 69.7)	REF			
**Sex**						
Male	45/97	46.4 (36.2, 56.8)	0.48 (0.30, 0.76)	0.002	0.31 (0.18, 0.54)	<0.001
Female	194/302	64.2 (58.5, 69.6)	REF			

***** n = Number positive for Crimean–Congo Hemorrhagic Fever Virus; N = Total sample examined; CI = Confidence Interval.

## Data Availability

The data presented in this study are available on request from the corresponding author.

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
