# Peer review of "Serological Prevalence of Crimean–Congo Hemorrhagic Fever Virus Infection in Small Ruminants and Cattle in The Gambia"

_pathogens, 2023, doi:10.3390/pathogens12060749_

Round 1

Reviewer 1 Report

 I feel this is a well-done study to clarify epidemiology of CCHF in the country. Please describe status of human infection in the country. Have you ever had a human case? It seems like there should be undetected human cases. Please propose a plan for human studies as this is the goal.

The main question is " Is there any evidence of CCHFV circulation in animals in the country?".

The conclusions are consistent. I have recommended that the you should discuss these results further in relationship with human cases in the country. You are presenting evidence of CCHFV circulation in animals, but what is the next step? Do you know of any confirmed human cases in the area? If not, you should propose a plan for looking for human cases. My conclusion from this result is that CCHFV human infections are undiagnosed or underdiagnosed in the area.  I think this is the main section and human infection needs to be clarified.

--

Author Response

REVIEWER 1  

Comments and Suggestions for Authors

 I feel this is a well-done study to clarify epidemiology of CCHF in the country. Please describe status of human infection in the country. Have you ever had a human case? It seems like there should be undetected human cases. Please propose a plan for human studies as this is the goal. 

The main question is " Is there any evidence of CCHFV circulation in animals in the country?". 

The conclusions are consistent. I have recommended that  you should discuss these results further in relationship with human cases in the country. You are presenting evidence of CCHFV circulation in animals, but what is the next step? Do you know of any confirmed human cases in the area? If not, you should propose a plan for looking for human cases. My conclusion from this result is that CCHFV human infections are undiagnosed or underdiagnosed in the area.  I think this is the main section and human infection needs to be clarified.

Response: We agree with the reviewer’s comments and have captured this more succinctly in the revised manuscript (lines 406 to 409). “Future studies that identify acute infections in at-risk human populations as well as febrile patients via detection of pathogen nucleic acid and /or IgM antibodies need to be conducted to demonstrate circulation of the virus and a more accurate risk for zoonotic disease transmission”.

Reviewer 2 Report

The study "Serological Prevalence of Crimean-Congo Hemorrhagic Fever Virus Infection in Small Ruminants and Cattle in The Gambia" focuses on the epidemiology of CCHFV in domesticated ruminants of Gambia which is a significant public and animal health concern. The study provides valuable information on the distribution of the CCHFV virus among livestock in different regions, which will be useful in developing targeted control measures and preventive strategies. Furthermore, it is important to monitor the prevalence of the virus in domestic animals to detect any emerging outbreaks and prevent the spread of the virus to humans. Overall, the findings of this study can help to inform public health policies, improve disease surveillance, and ultimately reduce the risk of CCHFV transmission to humans. Some of the serious major concerns are as follows:

Overall, the manuscript is well written and the topic is generally intriguing. The article contains few typographical and grammatical errors which need to be fixed before possible consideration.

Some of the concerns are as follows:

> Why did the authors choose to collect particular number of samples from a specific region? Did the authors apply the Proportional allocation method for this purpose? In that case, what was the total population in the targeted regions?

> Since, the study is funded by NBAF and WALIC, the authors could perform genomic detection assays as well as sequencing for more precise findings and in-depth analysis on CCHFV in the studied animals. Simple sero-prevalence data will not add up more to the global scientific community's understanding of CCHFV.

> Sero-prevalence results should be presented as significant or non-significant instead of higher or lower prevalence. 

> Line No. 18: Add p value after significant variation.

> In Introduction section, information on pathogenesis of CCHFV in the domesticated ruminants are missing.

> In Materials and Methods section (Line No. 68): Was it possible for the authors to collect so many blood samples and relevant data from the study population in just one month period?

> In Discussion section, the authors are advised to discuss the findings based on statistical analysis i.e., p value. The primary focus should be the discussion of significant results.

> There is no “conclusion section” at the end of the study!

--

Author Response

Reviewer 2

The study "Serological Prevalence of Crimean-Congo Hemorrhagic Fever Virus Infection in Small Ruminants and Cattle in The Gambia" focuses on the epidemiology of CCHFV in domesticated ruminants of Gambia which is a significant public and animal health concern. The study provides valuable information on the distribution of the CCHFV virus among livestock in different regions, which will be useful in developing targeted control measures and preventive strategies. Furthermore, it is important to monitor the prevalence of the virus in domestic animals to detect any emerging outbreaks and prevent the spread of the virus to humans. Overall, the findings of this study can help to inform public health policies, improve disease surveillance, and ultimately reduce the risk of CCHFV transmission to humans. Some of the serious major concerns are as follows: 

Overall, the manuscript is well written, and the topic is generally intriguing. The article contains few typographical and grammatical errors which need to be fixed before possible consideration.

Response: We thank the reviewer for highlighting this. We addressed the typo and grammatical errors in the revised manuscript.

Some of the concerns are as follows:

> Why did the authors choose to collect particular number of samples from a specific region? Did the authors apply the Proportional allocation method for this purpose? In that case, what was the total population in the targeted regions? 

Response: We thank the reviewer for the comments. This study was based on convenience sampling, which focused on collection of samples from target livestock populations, small ruminants, at livestock market holding facilities across the country, and the number of samples collected was based on the number of animals present at the market during the cross-sectional sampling period, and not on proportional allocation. A similar approach was adopted for sampling village cattle herds, which was a localized cross-sectional study in a cluster of villages in the Lower River Region. 

> Since, the study is funded by NBAF and WALIC, the authors could perform genomic detection assays as well as sequencing for more precise findings and in-depth analysis on CCHFV in the studied animals. Simple sero-prevalence data will not add up more to the global scientific community's understanding of CCHFV. 

Response: We appreciate the reviewer’s concern. However, we would like to mention that this represents the first study regarding the potential risk of CCHF in the Gambia which finding shall help inform health policy in The Gambia and guide further studies in at-risk human populations as described in the discussion. The current study primarily focused on an initial assessment to determine potential infection in target host species, domestic ruminants. That objective has been achieved and the results provide an indication of potential risk of disease to humans. A more in-depth study involving genetic characterization of the pathogen in animals and vectors, which is outside the scope of the current study, is planned to be undertaken in the future.    

> Sero-prevalence results should be presented as significant or non-significant instead of higher or lower prevalence.  

Response: We thank the reviewer for the comments. We did utilize the odds ratio and its significance to highlight the odds of getting infected for a susceptible host is higher or lower at various study sites and as a way of interpreting potential risk of zoonotic transmission. However, to address the reviewer’s concerns, we have presented “significant” differences in seroprevalence where appropriate in the results sections of the revised manuscript.

> Line No. 18: Add p value after significant variation.

Response: p value has been added.

> In Introduction section, information on pathogenesis of CCHFV in the domesticated ruminants are missing. 

We thank the reviewer for the comments. A statement on pathogenesis of CCHF in domestic ruminants is added in the revised manuscript (lines 50 to 54).

> In Materials and Methods section (Line No. 68): Was it possible for the authors to collect so many blood samples and relevant data from the study population in just one month period?

Response: We thank the reviewer. Yes, this was not only possible but very possible; and in fact, we have the possibility to collect more than 400 samples on a single day from both small ruminants and cattle. WALIC has a well-established network of Livestock Assistants/field assistants across the country in villages including close partnership with the Department of Livestock/Veterinary Services, The Gambia. This capacity is unique and allows sampling of large number of samples across the country within a very short period. This is a well planned and coordinated operation. See Faburay et al. 2005 for a similar example.  

> In Discussion section, the authors are advised to discuss the findings based on statistical analysis i.e., p value. The primary focus should be the discussion of significant results. 

Response: We thank the reviewer for the comments. We have included the P value/significance in the discussion comparing seroprevalence between regions and within the context of the potential risk for zoonotic transmission in the discussion section of the revised manuscript.

> There is no “conclusion section” at the end of the study!

Response: We thank the reviewer for the comments. A “conclusion section” has been added in the revised manuscript.

Reviewer 3 Report

Overall, the data presented is important in that it provides new epidemiological information from the previously unreported area. Even though there are previous publications from neighboring regions, the present data has value. I only have a couple of questions/suggestions: 

  • To give correct information, it is worth also mentioning GP38/GP85 and GP160 from the M segment (Line 43-introduction). 

  • The authors mention the possibility of misdiagnosing CCHFV symptoms. Could you please provide details regarding the case definition and diagnosis algorithm? (Line 346 in the discussion). 

  • I am curious if the authors have considered the inclusion of seroprevalence data in humans from neighboring countries that exhibit comparable seropositivity rates in small ruminants and cattle. 

  • In the discussion section, instead of solely mentioning similar seropositivity, it would be beneficial to include additional information to provide readers with a comprehensive understanding of the CCHFV seroprevalence in West Africa. Furthermore, the discussion could be enriched by including a couple more relevant studies such as PMID: 28719259 from Mali, PMID: 7573699 from Niger, and PMID: 37056704 from Cameroon.

Author Response

Reviewer 3

Comments and Suggestions for Authors

Overall, the data presented is important in that it provides new epidemiological information from the previously unreported area. Even though there are previous publications from neighboring regions, the present data has value. I only have a couple of questions/suggestions: 

To give correct information, it is worth also mentioning GP38/GP85 and GP160 from the M segment (Line 43-introduction). 

Response: We thank the reviewer for the comments. A statement on GP38/GP85 and GP160 is included in the revised manuscript.

The authors mention the possibility of misdiagnosing CCHFV symptoms. Could you please provide details regarding the case definition and diagnosis algorithm? (Line 346 in the discussion). 

Response: We thank the reviewer for the comments, the case definition of CCHF has been added in detail as requested in the revised manuscript (lines 371 to 373).

I am curious if the authors have considered the inclusion of seroprevalence data in humans from neighboring countries that exhibit comparable seropositivity rates in small ruminants and cattle. 

Response: We agree with the reviewer’s suggestion and have included the requested information/data in the revised manuscript (lines 313 to 315).

In the discussion section, instead of solely mentioning similar seropositivity, it would be beneficial to include additional information to provide readers with a comprehensive understanding of the CCHFV seroprevalence in West Africa. Furthermore, the discussion could be enriched by including a couple more relevant studies such as PMID: 28719259 from Mali, PMID: 7573699 from Niger, and PMID: 37056704 from Cameroon.

Response: We thank the reviewer for the comments. The requested information with corresponding citations/references have been provided in the revised manuscript (lines 354 to 358).

Round 2

Reviewer 2 Report

I commend the authors for their diligent efforts in enhancing the quality of the manuscript. The substantial improvements made have significantly strengthened the scientific validity of the article. Consequently, based on the revisions implemented, I confidently endorse the acceptance of the manuscript.

Overall, English quality is good. Minor editing of English language required